# Proton Range Measurement Precision in Ionoacoustic Experiments with Wavelet-Based Denoising Algorithm [note 1]

**DOI:** 10.3390/s25144247

**Published:** 2025-07-08

**Authors:** Elia Arturo Vallicelli, Andrea Baschirotto, Lorenzo Stevenazzi, Mattia Tambaro, Marcello De Matteis

**Affiliations:** Department of Physics, University and INFN Section of Milano–Bicocca, 20126 Milano, Italy; andrea.baschirotto@unimib.it (A.B.); l.stevenazzi1@campus.unimib.it (L.S.); mattia.tambaro@unimib.it (M.T.); marcello.dematteis@unimib.it (M.D.M.)

**Keywords:** circuits and systems for biomedical applications, radiation therapy, ultrasound sensors

## Abstract

This work presents the experimental results of a wavelet transform denoising algorithm (WTDA) that improves the ionoacoustic signal-to-noise ratio (SNR) and proton range measurement precision. Ionoacoustic detectors exploit the ultrasound signal generated by pulsed proton beams in energy absorbers (water or the human body) to localize the energy deposition with sub-millimeter precision, with interesting applications in beam monitoring during oncological hadron therapy treatments. By improving SNR and measurement precision, the WTDA allows significant reduction of the extra dose necessary for beam characterization. To validate the WTDA’s performance, two scenarios are presented. First, the WTDA was applied to the ionoacoustic signals from a 20 MeV proton beam, where it allowed for increasing the SNR by 17 dB and improving measurement precision by a factor of two. Then, the WTDA was applied to the simulated signals from a 200 MeV clinical beam where, compared to state-of-the-art algorithms, it achieved a −80% dose reduction when achieving the same 30 μm precision and a six-fold precision improvement for the same 17 Gy dose deposition.

## 1. Introduction

Oncological hadron therapy treatments exploit the energy deposition curve of protons and ions (called the Bragg curve, Figure 1), which concentrates the radiation damage on the tumor volume while sparing the surrounding healthy tissues [1]. Such localized dose deposition, however, requires precise monitoring to maximize treatment efficacy and minimize collateral damage to the tissues surrounding the treatment volume. An emerging beam monitoring technique exploits the thermo-induced acoustic pulse generated by rapid energy deposition for localization of the Bragg peak with sub-millimeter precision, compared to the few-millimeter precision of traditional techniques (PET and nuclear imaging techniques) [2,3,4,5,6,7,8,9,10,11,12,13,14,15,16,17,18,19,20,21,22,23,24]. Ionoacoustic detectors measure the acoustic wave time of flight (ToF) to calculate the position of the BP in the absorber with a precision of a few tens of microns in pre-clinical scenarios and a few hundred microns in clinical scenarios. However, clinical scenarios are particularly critical in terms of signal-to-noise ratio (SNR) because the dose depositions involved in clinical treatments are very low, typically a few mGy per beam pulse. The amplitude of the acoustic signal is proportional to the dose deposited during the pulse, and this results in a poor SNR (below 0 dB) that needs to be improved in post-processing to reject random noise fluctuations and achieve the desired measurement precision.

The typical approach used in state-of-the-art experiments is to average the signals generated by several (up to thousands of) beam pulses, improving the SNR, following Equation (1), where SNR_avg_ is the final SNR after averaging the signals from N_p_ pulses and SNR_1p_ is the single-pulse SNR.(1)$${SNR}_{avg}={SNR}_{1p}+10\log_{10}N_{p}$$

However, this approach involves an additional dose delivered to the patient for each pulse of the beam and is therefore not compatible with clinical treatments. Figure 2 shows this extra-dose deposition in a typical clinical scenario with 10 mGy/pulse. Typical state-of-the-art experiments require increasing the single-pulse SNR by 20 to 30 dB, requiring averaging of 100–1000 different pulses and thus resulting in an additional dose of several Gy. To overcome this limitation, this paper presents the application of a denoising algorithm based on wavelet transform that allows one to reject noise and improve the SNR on a single signal (single-beam pulse) without requiring averaging and the related extra dose.

The performance of the hereby proposed wavelet transform denoising algorithm (WTDA) is first evaluated in a pre-clinical experimental scenario (20 MeV protons), described in Section 2 with reference to [25,26]. The denoising algorithm is described in Section 2 and its performance is evaluated in Section 3 in terms of SNR improvement, measurement precision improvement and additional dose reduction compared to averaging. Then, in Section 4 the impact of such a denoising approach on a simulated clinical scenario in terms of dose reduction and precision improvement is shown. Finally, in Section 5, conclusions will be drawn.

## 2. Materials and Methods

### 2.1. Ionoacoustic Experimental Setup

The typical ionoacoustic experimental setup is composed of a proton beam and a water tank as an energy absorber. With reference to Figure 2 and [10], the experimental setup consists of a 20 MeV proton beam incident in a water phantom (energy absorber). The hereby described experiment was carried out at the Maier-Leibniz Laboratory Tandem accelerator in Garching, Munich. The Bragg peak is 4.1 mm from the water tank entrance foil. The beam deposits 10^6^ protons/pulse and has a diameter of 2 mm. For each pulse a dose of 0.8 Gy is deposited at the Bragg peak, generating a localized pressure increase of 80 Pa. The main parameters of the experimental setup are summarized in Table 1.

The ionoacoustic signal propagates in the water until it reaches a piezoelectric sensor placed at 25.6 mm in the axial direction (Videoscan V311, Olympus, Tokyo, Japan). The pressure signal at the sensor surface is 5 Pa due to spherical attenuation and has a frequency of 2.3 MHz. The acoustic sensor has a sensitivity of 10 µV/Pa and is connected to an analog front-end consisting of a low noise amplifier (LNA), a low-pass filter and an analog-to-digital converter. The LNA has two stages with a total of 80 dB in-band gain, an equivalent input noise of 0.9 nV/√Hz and a 0.6 dB noise figure [10].

The low-pass filter is a Rauch (2nd order) with a cut-off frequency of −3 dB at 4.5 MHz to reject out-of-band noise and act as an anti-aliasing filter for the ADC (10-bit, 80 MS/sec). Finally, an FPGA takes care of automatically acquiring the data and transferring them to a server via USB where they are processed to obtain information on the physical phenomenon and locate the BP.

This setup is described in more detail in [10], and the main characteristics of the detector are listed in Table 2. The ionoacoustic signal is shown in Figure 3, where the dotted line signal represents the single-pulse 11 dB SNR signal and the black line signal represents a 1000-fold averaged signal, with a final SNR of 42 dB. The time of flight is measured from the energy deposition (marked by a trigger signal generated by the accelerator electronics) and the peak of the acoustic wave and is equal to 17.1 μs, corresponding to a BP–sensor distance of 25.6 mm.

### 2.2. Wavelet Denoising

The single-pulse signal was processed using the fast wavelet transform denoising algorithm proposed by Mallat and based on the sym6 wavelet family, an empirical Bayesian method with a Cauchy prior, median thresholding and level-dependent noise estimate [25,26]. These parameters have been chosen empirically over other solutions since they provided a better noise rejection and serve as a proof-of-concept of the application of wavelet denoising in ionoacoustics, whereas a complete optimization study is needed to find the best tradeoff between denoising performance and computational costs. Figure 4 shows the generic block-diagram of the algorithm. The time-domain signal is decomposed using the wavelet transform in 14 different levels, each corresponding to a different bandwidth. In particular, level *n* corresponds to the bandwidth from f_N_/2_n_ to f_N_/*n*, where f_N_ is the Nyquist frequency (40 MHz in this setup). Then, for each level, the median of the signal is calculated and the signal is compared to a threshold to discriminate the deterministic signal from random noise fluctuations. After thresholding, the output denoised signal is obtained by adding the denoised signals from each level. To evaluate the WTDA’s performance in terms of SNR increase, the SNR of the input and output signals and the signal power are calculated by measuring the 0-peak signal amplitude, and the noise power is estimated by calculating the standard deviation of the signal in a time interval of 1024 samples just before the beam trigger, to avoid interferences from the accelerator electronics and the electromagnetic pulse associated with the beam generation. Such interferences occur during the beam pulse (t = 0) but are not superimposed onto the ionoacoustic signal thanks to the relatively long (17 μs) acoustic wave time of flight from the Bragg peak to the acoustic sensor. The measurement precision is estimated by repeating the BP location measurement (that is, measuring the ToF and calculating the BP–sensor distance by multiplying for the speed of sound in water) for 8000 different beam pulses. Random noise fluctuations result in a slightly different ToF measurement compared to the true value. The measurement precision is obtained by calculating the standard deviation of the BP position measured for 8000 beam pulses.

## 3. Experimental Results—20 MeV Pre-Clinical Scenario

### 3.1. Signal-to-Noise Ratio Enhancement

Figure 5 shows the time-domain signal before and after denoising.
Figure 5—top shows the 13 dB SNR signal acquired by the ProSD AFE that was used as an input signal for both the WTDA algorithm and an averaging algorithm for reference. This signal was generated by a 0.8 Gy dose deposition.Figure 5—middle shows an 80-fold averaged signal (averaging 80 different time-domain signals with 11 dB SNR, acquired during different beam pulses). The SNR has improved to 30 dB according to Equation (1) and the total dose deposition is 64 Gy, 80 times the single-pulse deposition.Figure 5—bottom shows the output of the WTDA signal, characterized by the same 30 dB SNR as the 80-fold average signal, but obtained from a single-pulse signal with a total dose deposition of 0.8 Gy.

In this scenario, the WTDA allows for a 17 dB SNR increase that, however, comes without any extra dose, compared to a 63.2 Gy extra dose required by simple time-domain averaging. The efficacy in SNR improvement of the WTDA was characterized by evaluating the output SNR as a function of the input SNR. A reference (almost noise-free) signal was generated by averaging the signals from 8000 different beam pulses. Then noise power (with the same bandwidth as the analog front-end) was added to achieve a set of signals with different SNR values. The WTDA was applied to these signals and the output SNR was measured.

The results are shown in Figure 6, where the reference dotted line represents the scenario where SNRout is equal to SNRin and marks the minimum SNRin value for the WTDA to be effective. In particular, when the input signal SNR is more than 10 dB the WTDA allows for an increase in SNR, whereas for SNR values less than 10 dB the WTDA cannot distinguish the signal from noise and this results in a loss of SNR. The highest SNR improvement is obtained for SNRin above 15 dB, where an SNR increase of up to 26 dB is observed.

Finally, Table 3 summarizes the performance of this work compared to state-of-the-art ionoacoustic experiments at 20 MeV energy.

### 3.2. Measurement Precision Improvement

The capability of the WTDA to improve measurement precision was evaluated by repeating the BP position measurement and by calculating the standard deviation of the resulting value. The measurement precision, defined as the standard deviation of the BP location, was then evaluated for different input SNRs comparing averaging and the WTDA. The results are shown in Figure 7, where the dotted line represents the precision allowed by a signal with a certain SNR obtained by averaging multiple beam pulses and the black line represents the precision obtained by the WTDA for a given SNRin value. Figure 7 shows that whereas the WTDA allows for an improvement in SNR for SNRin values above 10 dB, this SNR improvement results in a precision improvement only for SNRin values above 20 dB. For example, an input signal with a 15 dB SNR allows 11 μm precision, but after applying the WTDA this precision worsens to 22 μm. On the other hand, an input signal with 24 dB SNR results in 5 μm precision that improves to 1 μm after applying the WTDA. To achieve the same precision without the WTDA, the input signal SNR would have been equal to 29 dB. This characterization allows one to define the best working spot of the WTDA for precision enhancement, which is with 22–24 dB input SNR.

In reference [10], three different scenarios are presented, corresponding to the single-pulse signal and 10-fold and 100-fold average. These scenarios are reported in Table 4 for reference. The 10-pulse scenario has a 22 dB SNR and thus fits well with the WTDA requirements. The output precision is equal to 4 μm, compared to the 7.5 μm reported in [10]. To achieve the same precision without the WTDA, an input SNR of 25 dB would have been required. However, to achieve a 25 dB SNR without the WTDA, 10 additional averages are needed, requiring a total dose of 16 Gy instead of 8 Gy. Thus, in this scenario the WTDA allows for a 50% dose reduction.

## 4. Simulation Results—200 MeV Clinical Scenario

The most promising application of the presented ionoacoustic WTDA is in clinical scenarios, where currently one of the main limiting factors of the ionoacoustic technique is the high doses required to localize the BP. The most critical clinical scenarios are the ones with higher beam energy because the large associated BP volume results in very low dose depositions, low signal amplitudes and, thus, low SNR [19]. In this section, the WTDA has been applied to a simulated ionoacoustic signal generated by a 200 MeV proton beam. These values represent the experimental setup described in [19]. Geant4 [27] has been used to achieve the 3D dose deposition profile (Figure 8) and the Bragg curve, which is characterized by a beam range of 25.9 cm and a BP_FWHM_ of 20.1 mm and 35 mGy dose deposition at the BP (compared to the 4.1 mm range, 320 μm BP_FWHM_ and 0.8 Gy at the BP for the 20 MeV scenario described in Section 2). This leads to a lower signal amplitude and a −2 dB single-pulse SNR. The pressure increase was calculated from the 3D dose deposition map and was used as the input of a Matlab k-Wave model to simulate acoustic wave propagation in water [28]. A sensing point was placed 5 cm from the BP in the axial direction (the same location as in [19]). The acquired acoustic signal was processed by a dedicated Matlab script to model sensor and electronic noise power and bandwidth. The single-pulse signal has an SNR of −2 dB. Averaging was applied to improve the SNR to 21 dB (200-fold average for a total dose of 7 Gy) which is the value where both averaging and the WTDA achieve the same precision value.

The 21 dB SNR signal is shown in Figure 9 (black line) along with the noise-free signal (dotted line) for reference.

Then, the WTDA was applied in different scenarios, applying first averaging (to improve the SNR from −2 dB into the optimal WTDA working range of 21–25 dB SNR, that is, from 200- to 500-fold averaging) and then the WTDA. The averaging + WTDA results were compared with averaging alone for the same final precision. These results are shown in Figure 10. In the 21–25 dB SNR range, a small improvement in SNR results in a large precision improvement compared to averaging, as shown in Figure 7 where the WTDA line has a steeper slope than the averaging line. This characteristic can be found in Figure 10 as well, where the WTDA allows for a large improvement in precision for a small additional dose compared to averaging. In particular, Figure 10—top shows that for a given dose deposition (17 Gy, equal to 500 beam pulses and an SNR of 25 dB after averaging) the WTDA grants better precision, with 80% lower sigma compared to averaging (30 μm vs. 180 μm). Moreover, to achieve the same 30 μm precision only by averaging, a 62 Gy dose deposition (1770 beam pulses) is required compared to the 17 Gy dose required by the WTDA, resulting in a 70% dose reduction as shown in Figure 10—bottom. Whereas 17 Gy is higher than typical treatment fractions, this value can be greatly reduced by improving the sensor and analog stages, thus lowering the detector noise power and achieving the same SNR in this scenario with lower total doses.

Figure 11 summarizes the dose deposition reduction compared to averaging to achieve a given relative error compared to the BPFWHM size.

Figure 12 shows the histograms of the measured BP position for averaging (500-fold) and the WTDA, highlighting the 6-fold improvement in measurement precision allowed by the WTDA.

Finally, Table 5 reports a performance summary of the proposed WTDA signal processing and the state-of-the-art averaging approach in the experimental scenario reported in [19].

## 5. Conclusions

This paper shows how the application of a denoising algorithm based on wavelet transform allows improvement of the signal-to-noise ratio and proton range measurement precision in ionoacoustic detectors without the need for an extra dose associated with time-domain averaging. The most interesting application of ionoacoustic detectors is in the field of oncological hadron therapy, but they are currently limited by the high dose necessary to obtain a signal of sufficient SNR to localize the Bragg peak. All major state-of-the-art experiments average hundreds or thousands of pulses to improve the SNR, but this entails a relevant extra dose given the poor denoising efficiency of averaging. The results presented in this paper show how the use of advanced denoising algorithms can drastically reduce the dose needed to locate the Bragg peak, potentially bringing this technique closer to clinical application. This paper is part of an all-around improvement work on detector technology, optimizing the acoustic sensor, the electronics and the signal processing. Switching from general-purpose detectors to dedicated detectors that exploit multi-channel sensors, integrated electronics and advanced digital signal processing is essential for clinical applications of the ionoacoustic technique.

## Figures and Tables

**Figure 1 sensors-25-04247-f001:**
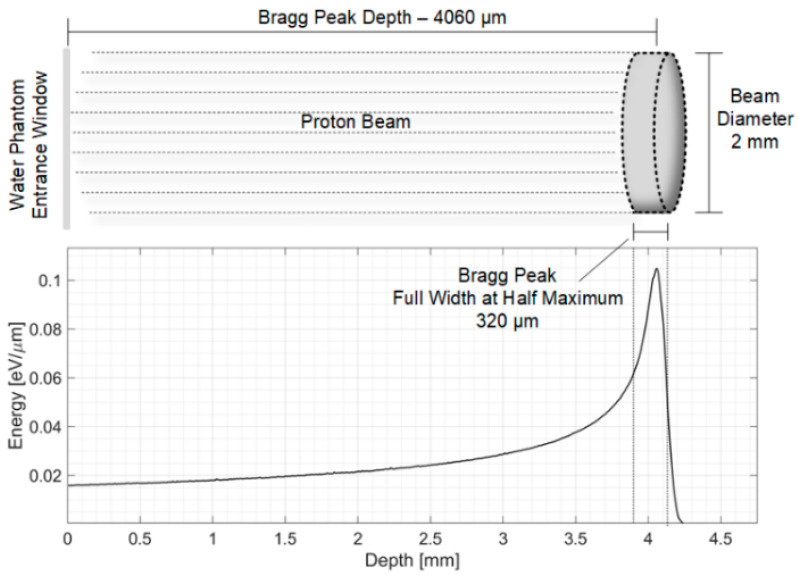
Bragg curve for 20 MeV protons.

**Figure 2 sensors-25-04247-f002:**
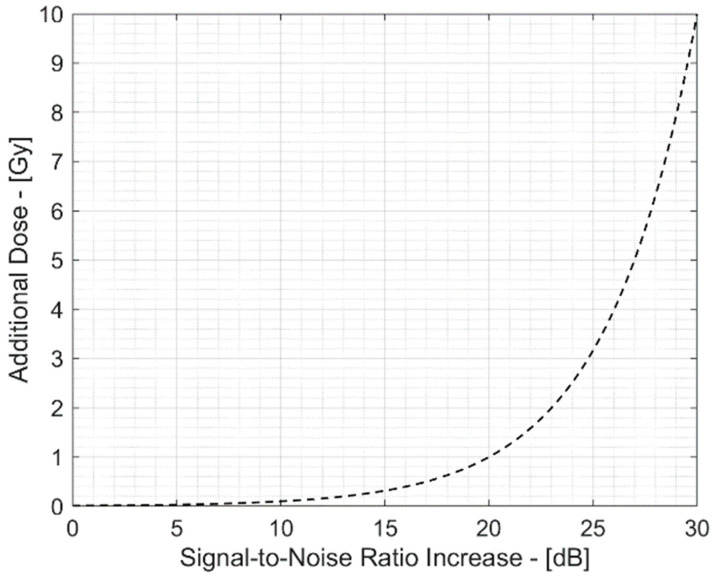
Extra dose due to SNR increase by multiple beam pulse averaging.

**Figure 3 sensors-25-04247-f003:**
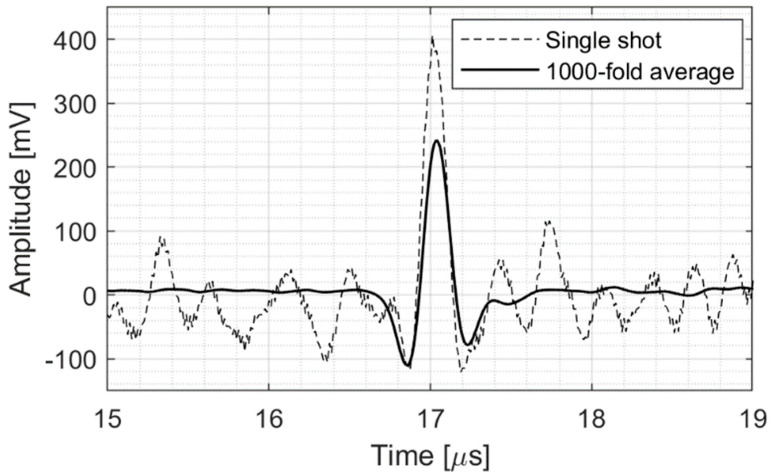
Single−pulse and 1000-fold average ionoacoustic signals.

**Figure 4 sensors-25-04247-f004:**
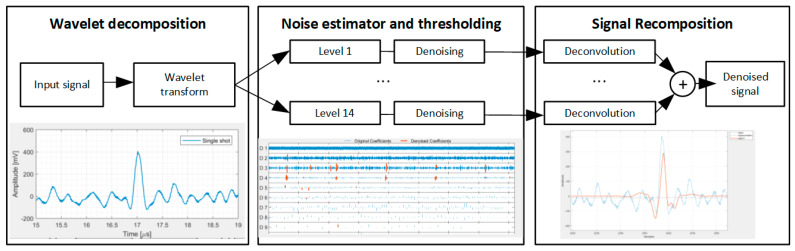
Block scheme of the wavelet denoising algorithm.

**Figure 5 sensors-25-04247-f005:**
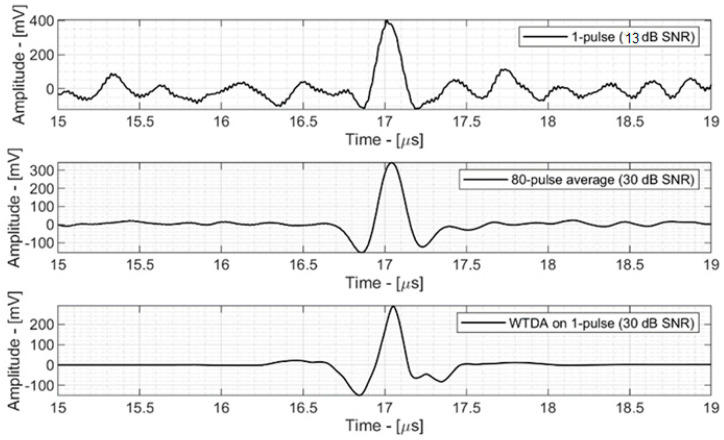
Single−pulse signal, 80-fold signal and single-pulse signal processed with WTDA.

**Figure 6 sensors-25-04247-f006:**
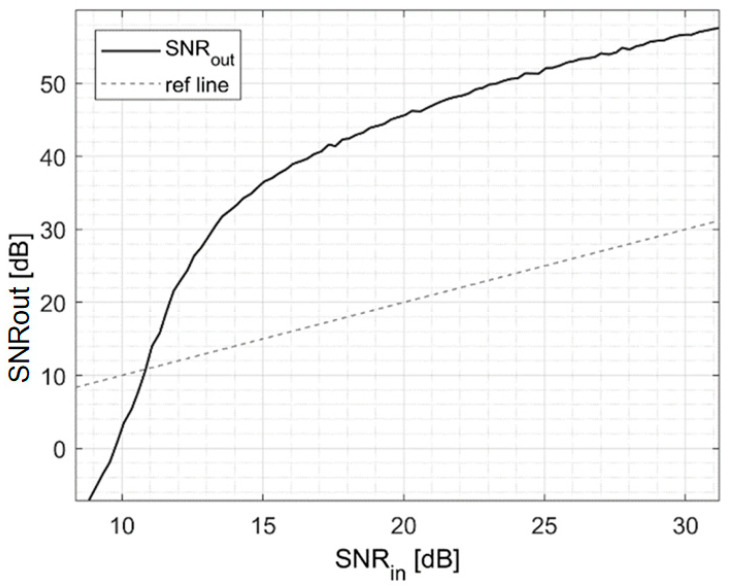
WTDA output SNR as a function of input SNR.

**Figure 7 sensors-25-04247-f007:**
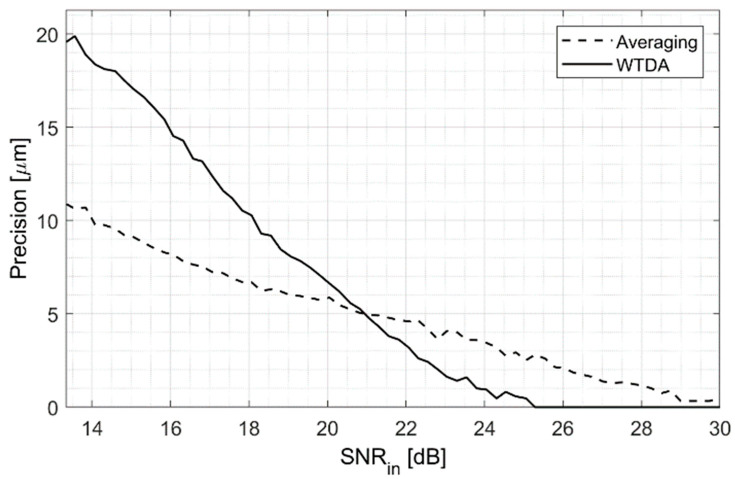
The 20 MeV proton range measurement precision for averaging and the WTDA as a function of input SNR.

**Figure 8 sensors-25-04247-f008:**
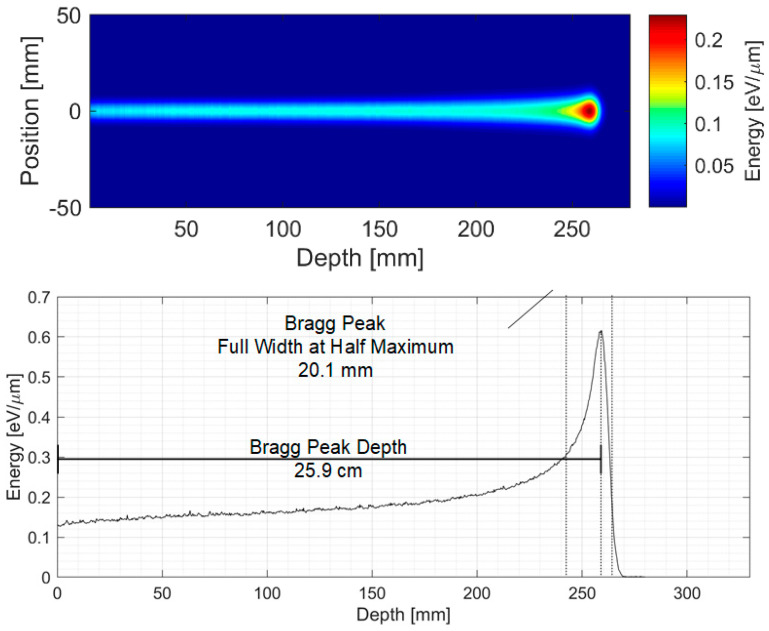
The 200 MeV proton 2D dose deposition profile and Bragg curve.

**Figure 9 sensors-25-04247-f009:**
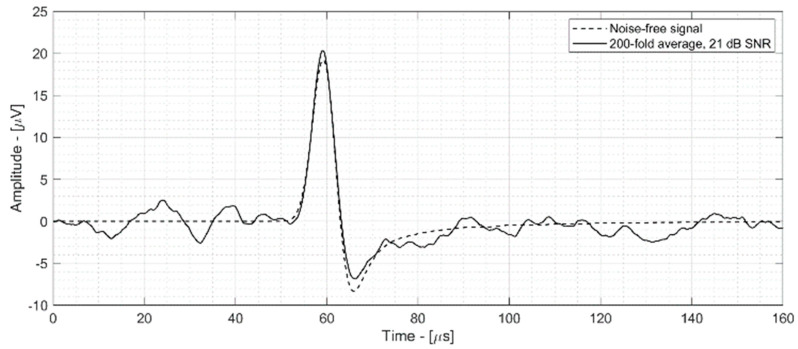
Simulated 200 Mev proton beam ionoacoustic signal with (black) and without (dotted) sensor and electronic noise.

**Figure 10 sensors-25-04247-f010:**
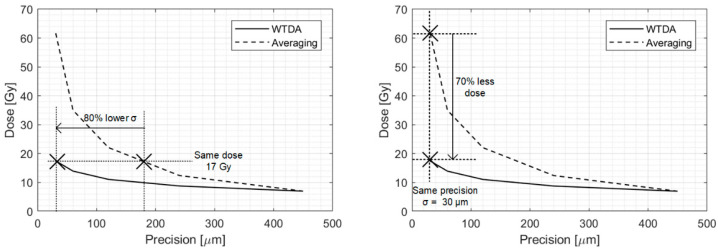
Dose to achieve a given measurement precision for the WTDA and averaging.

**Figure 11 sensors-25-04247-f011:**
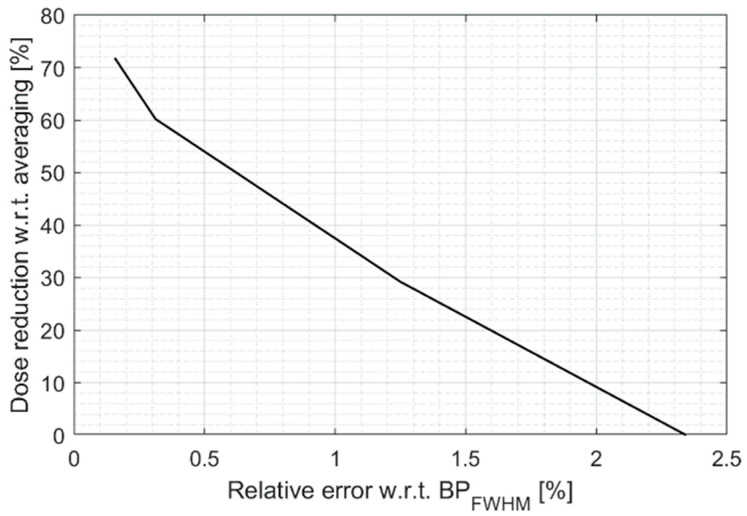
Dose reduction allowed by wavelet denoising as a function of measurement precision expressed as a fraction of the Bragg peak size.

**Figure 12 sensors-25-04247-f012:**
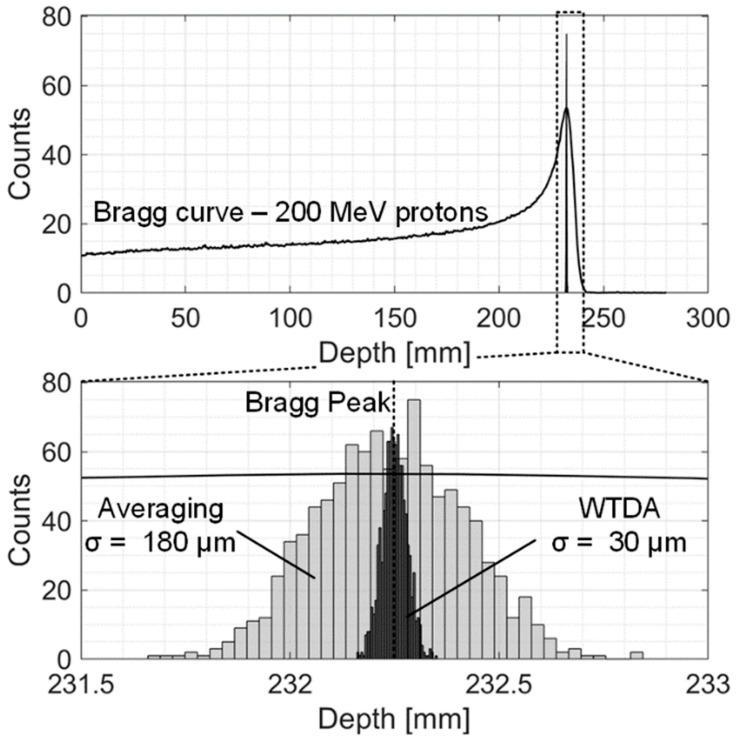
Histograms of the measured BP location (repeated 1000 times) for the WTDA and averaging.

**Table 1 sensors-25-04247-t001:** Experimental setup parameters.

Parameter	Symbol	Value
Particle Energy	E	20 MeV
Current Pulse Time Window	TW	120 ns
Stress confinement	t_stress_	220 ns
Proton Bunch Number	N	1 M
Beam Diameter	D_B_	2 mm
Pulse Equivalent Injected Charge	Q_IN_	0.16 fC
Deposited Energy Dose	E_DEP_	0.8 Gy
BP FWHM	BP_FWHM_	0.32 mm
Beam Depth Range	R	4.060 mm
BP Volume	BP_VOLUME_	0.64 mm^3^
Sound Speed in Water (22.3 °C)	c_w_	1492 m/s

**Table 2 sensors-25-04247-t002:** Main AFE parameters.

	Parameter	Value
**Acoustic Sensor**	Pass-band Sensitivity	~10 µV/Pa
	Pass-band Frequency Range (− 6 dB)	2.45–4.95 MHz
	Capacitance	1 nF
	Output Noise Power	5.82 µV_RMS_
	Input Referred Noise	~580 mPa_RMS_
**LNA**	Pass-band Gain	60–80 dB
	−3dB lower frequency	10 kHz
	−3dB upper frequency	4 MHz
	Input Referred Noise Voltage	2.25 µV_RMS_
**Low-Pass Filter**	Pass-Band Gain	0 dB
	Pole Frequency	4 MHz
	Pole Quality Factor	0.707
**A-to-D Converter**	Sampling Frequency	80 MHz
	Number of Bits	10
	Equivalent Number of Bits (ENOB)	9.5

**Table 3 sensors-25-04247-t003:** Performance summary with respect to the state of the art—SNR.

Parameter	Assmann et al., 2015 [8]	Vallicelli et al., 2020 [10]	This Work
Proton Energy	20 MeV	20 MeV	20 MeV
Detector Features	Commercial Front-end	Dedicated LNA	Dedicated LNA
DSP	Averaging	Averaging	WTDA
SNR for 1 pulse (0.8 Gy)	6 dB	13 dB	30 dB
Dose to achieve 30 dB SNR	200 Gy	64 Gy	0.8 Gy

**Table 4 sensors-25-04247-t004:** Performance summary with respect to the state of the art—precision.

Parameter	Single Pulse [11]	10-Pulse Average [11]	20-Pulse Average [11]	10-Pulse Average + WTDA
SNR	13 dB	22 dB	25 dB	51 dB
Dose	0.8 Gy	8 Gy	16 Gy	8 Gy
BP position	4073 μm	4073 μm	4073 μm	4073 μm
Precision	21 μm	7.5 μm	4 μm	4 μm
Relative Error (%)	0.5 %	0.2 %	0.1%	0.1%

**Table 5 sensors-25-04247-t005:** Performance summary with respect to the state of the art—200 MeV proton clinical scenario.

Parameter	[10] Averaging	This Work: 500-Pulse Average + WTDA
Single-pulse SNR	−2 dB	−2 dB
500-pulse SNR	21 dB	21 dB
Dose per pulse	35 mGy	35 mGy
Npulse to achieve 30 μm precision	1770	500
Dose to achieve 30 μm precision	62 Gy	17 Gy
Precision with 17 Gy dose	180 μm	30 μm

## Data Availability

The raw data supporting the conclusions of this article will be made available by the authors on request.

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
