# Peer review of "Proton Range Measurement Precision in Ionoacoustic Experiments with Wavelet-Based Denoising Algorithm†"

_sensors, 2025, doi:10.3390/s25144247_

Round 1

Reviewer 1 Report

Comments and Suggestions for Authors

The study presents a wavelet transform denoising algorithm (WTDA) that improves the signal-to-noise ratio (SNR) of ionoacoustic signals by up to 17 dB in a 20 MeV proton beam setup and achieves a 6-fold precision enhancement (30 μm vs. 180 μm) in simulated 200 MeV clinical scenarios. Notably, WTDA reduces the required dose for Bragg peak (BP) localization by 70–80% compared to conventional averaging methods. These results address a critical limitation in clinical ionoacoustic monitoring by minimizing radiation exposure during proton therapy. The work builds on prior studies (e.g., Assmann et al., 2015; Lehrack et al., 2017) that rely on pulse averaging for SNR improvement. While wavelet denoising has been explored in other signal processing contexts, its application to ionoacoustics for dose reduction is novel. 

However, several methodological and presentation issues require clarification before publication.
1)Though the 200 MeV Geant4/k-Wave simulation aligns with realistic scenarios,  experimental validation in clinical settings is absent. The 200 MeV simulation assumes ideal water phantoms. Real-world factors (e.g., tissue interfaces, motion artifacts) are ignored, limiting clinical relevance. Tissue heterogeneity and acoustic attenuation in human tissues are not modeled, potentially overestimating clinical performance. Cross-validation with independent datasets or phantom experiments is needed.

2)While the wavelet method is described, critical details (e.g., thresholding criteria, noise estimation in pre-trigger intervals) are insufficient for replication. SNR calculation relies on pre-trigger noise estimation, which may not capture non-stationary noise during beam operation. The noise is assumed Gaussian—is this valid given beam-induced electromagnetic interference? Real-world clinical noise may include non-stationary artifacts (e.g., patient motion). Please justify the reason for wavelet choice (sym6)—why not other wavelets (e.g., Daubechies, Haar)? The paper uses median thresholding with a Cauchy prior—why this choice? How does thresholding affect signal distortion?

3)Benchmark against other denoising methods (e.g., Kalman filters, deep learning) to contextualize WTDA’s advantages.

4)Clarify if the proposed 17 Gy dose (Fig. 10) aligns with clinical safety standards (typical fractions are 1–2 Gy).

5)Writing improvement: 
oFig. 6 lacks Y-axis label.
o"μsing" (p.5, Sec. 2.2) → "using"
oMissing references for Geant4 & k-Wave in Methods.

Author Response

1)Though the 200 MeV Geant4/k-Wave simulation aligns with realistic scenarios,  experimental validation in clinical settings is absent. The 200 MeV simulation assumes ideal water phantoms. Real-world factors (e.g., tissue interfaces, motion artifacts) are ignored, limiting clinical relevance. Tissue heterogeneity and acoustic attenuation in human tissues are not modeled, potentially overestimating clinical performance. Cross-validation with independent datasets or phantom experiments is needed.

2)While the wavelet method is described, critical details (e.g., thresholding criteria, noise estimation in pre-trigger intervals) are insufficient for replication. SNR calculation relies on pre-trigger noise estimation, which may not capture non-stationary noise during beam operation. The noise is assumed Gaussian—is this valid given beam-induced electromagnetic interference? Real-world clinical noise may include non-stationary artifacts (e.g., patient motion). Please justify the reason for wavelet choice (sym6)—why not other wavelets (e.g., Daubechies, Haar)? The paper uses median thresholding with a Cauchy prior—why this choice? How does thresholding affect signal distortion?

3)Benchmark against other denoising methods (e.g., Kalman filters, deep learning) to contextualize WTDA’s advantages.

A:  All these remarks are true. This paper focuses on showing how advanced DSP algorithms such as wavelet denoising can improve SNR in ionoacoustics, where the standard algorithm in state of the art is just time domain averaging. Ionoacoustic detectors are very immature in terms of technology development and this paper aims to highlight the impact that a dedicated approach could have in terms of dose reduction, focusing on two case of study and comparing the impact of  a standard wavelet denoising approach vs. averaging. A comment about this has been added in the text. Obviously, in a real-world scenario the impact of patient tissues limits the amplitude of the recorded acoustic signal and thus the SNR. Gaussian noise is however a good approximation (confirmed by FFT analysis of noise in experimental scenarios) because the EM pulse generated by the accelerator is temporally separated from the ionoacoustic signal due to the acoustic wave time of flight and thus can be easily discarded. 

4)Clarify if the proposed 17 Gy dose (Fig. 10) aligns with clinical safety standards (typical fractions are 1–2 Gy).

A: We added a comment about this. 17 Gy in this study is limited by the state of the art detector performance that we are simulating. Such detector is wide band, single channel and non optimized, thus the ouput SNR is relatively low. Improving the analog stages technology (e.g. high sensitivity pvdf sensors, dedicated low-noise amplifiers and multichannel detectors) can lower detector noise power and achieve the same SNR of this study with significantly less dose.

5)Writing improvement: 
oFig. 6 lacks Y-axis label.
o"μsing" (p.5, Sec. 2.2) → "using"
oMissing references for Geant4 & k-Wave in Methods.

A: Fixed

Reviewer 2 Report

Comments and Suggestions for Authors

In this paper, the authors developed and described a wavelet transform noise reduction algorithm (WTDA). This algorithm is designed to improve the signal-to-noise ratio (SNR) and measurement accuracy of ion-acoustic detectors used for monitoring. The main focus is on reducing the additional dose required for accurate Bragg peak (BP) localization, which addresses a critical limitation of current ion-acoustic methods in the clinical setting.
By processing single-pulse signals, WTDA eliminates the need for extensive pulse averaging, significantly reducing the additional dose delivered to patients. This is an important step forward for clinical applications where dose minimization is critical.

The authors' results bridge the gap between preclinical and clinical ion-acoustic monitoring, offering a viable solution for highly accurate, real-time proton range verification in therapy.

Despite the above shortcomings, I would like to draw the authors' attention to the following areas for improvement:
1)The performance of WTDA is limited to input SNR values above 10 dB. At very low SNR (<10 dB), the algorithm cannot distinguish signal from noise, potentially reducing accuracy. This limits its use in extremely low-dose scenarios without pre-averaging.

2)The algorithm's dependence on multi-scale wavelet transforms and empirical Bayesian methods may increase computational complexity, posing challenges for real-time implementation in clinical systems.

3)While the 20 MeV results are experimentally validated, the 200 MeV scenario is based on simulations. Clinical validation is needed to confirm the algorithm's performance under realistic conditions, including tissue heterogeneity and noise variability.

4)he study assumes the use of specialized detectors and optimized analog front-end electronics (e.g., high-sensitivity sensors, low-noise amplifiers). Adaptation of WTDA to existing clinical equipment may require additional modifications.

Author Response

1)The performance of WTDA is limited to input SNR values above 10 dB. At very low SNR (<10 dB), the algorithm cannot distinguish signal from noise, potentially reducing accuracy. This limits its use in extremely low-dose scenarios without pre-averaging.

A: It is correct. However, in ultra-low doses scenario the SNR can be improved without pre-averaging also leveraging on multichannel acquisition and beamforming algorithms that lowers random sensor noise (each channel noise is independent and is averaged across the multichannel array) while preserving the deterministic acoustic signal.  

2)The algorithm's dependence on multi-scale wavelet transforms and empirical Bayesian methods may increase computational complexity, posing challenges for real-time implementation in clinical systems.

A: We agree on this point. However, most current dosimetry applications in hadron therapy are not performed online and in real time. 

3)While the 20 MeV results are experimentally validated, the 200 MeV scenario is based on simulations. Clinical validation is needed to confirm the algorithm's performance under realistic conditions, including tissue heterogeneity and noise variability.

4)he study assumes the use of specialized detectors and optimized analog front-end electronics (e.g., high-sensitivity sensors, low-noise amplifiers). Adaptation of WTDA to existing clinical equipment may require additional modifications.

A: Again, we completely agree. This paper focuses solely on showing how an improvement on the denoising DSP stages can improve the detector overall efficacy in determining the Bragg peak position. Real-life conditions such as patient inhomogeneities and detector positioning are not considered in this study, but need to be addressed to apply this technique in clinical scenarios.  

We added some clarifications in the manuscript to highlight these points.

Reviewer 3 Report

Comments and Suggestions for Authors

please,  place the tables (for example table 1) in section 2, which is where it belongs?

on page 3, line 83 it appears for the first time in the LNA document, but its definition is not found

In that same paragraph, they refer to a "low-pass filter" and in the next paragraph they refer to the same filter as "low pass", please write it in the same way as many times as necessary.

On line 112 it says μsing, is that correct?

on line 127 - 128, 171, 273, 284.. he misspelled the word ion-acoustic or ionoacoustic?? same in Figure 9 

  • Ionoacousticsis a field that utilizes the acoustic waves generated by the interaction of ion beams with tissue or other materials.  
  • Ion-acoustic wavesare a specific type of wave that can exist in plasmas. They are not directly related to the pulsed ion beam energy deposition process that characterizes ionoacoustics.  
  • In the context of medical imaging and therapy, ionoacoustics is used for tasks like range verification in proton therapy, where the depth and distribution of energy deposition are determined from the generated acoustic signals.

on line 170, 199, 265, reference is made to Table III,Table IV, Table V and that tables does not exists

please correct author contributions.

Is reference 28 correct?

Author Response

Thank you, all these typos have been corrected.

Round 2

Reviewer 1 Report

Comments and Suggestions for Authors

Authors have given the explanation about my concerns.